# Telmisartan Loaded Nanofibers Enhance Re-Endothelialization and Inhibit Neointimal Hyperplasia

**DOI:** 10.3390/pharmaceutics13111756

**Published:** 2021-10-21

**Authors:** Chen-Hung Lee, Kuo-Sheng Liu, Julien George Roth, Kuo-Chun Hung, Yen-Wei Liu, Shin-Huei Wang, Chi-Ching Kuo, Shih-Jung Liu

**Affiliations:** 1Division of Cardiology, Department of Internal Medicine, Chang Gung Memorial Hospital-Linkou, Chang Gung University College of Medicine, Taoyuan 33305, Taiwan; 0062@cgmh.org.tw (K.-C.H.); anglevvings@gmail.com (Y.-W.L.); s22012985@gmail.com (S.-H.W.); 2Department of Thoracic and Cardiovascular Surgery, Chang Gung Memorial Hospital, Taoyuan 33305, Taiwan; liuks@cgmh.org.tw; 3Institute for Stem Cell Biology & Regenerative Medicine, Stanford University School of Medicine, Stanford, CA 94305, USA; jgroth@stanford.edu; 4Research and Development Center of Smart Textile Technology, Institute of Organic and Polymeric Materials, National Taipei University of Technology, Taipei 10608, Taiwan; 5Department of Mechanical Engineering, Chang Gung University, Taoyuan 33302, Taiwan; 6Department of Orthopedic Surgery, Chang Gung Memorial Hospital-Linkou, Taoyuan 33305, Taiwan

**Keywords:** drug-loaded nanofibers, telmisartan, PLGA, controlled release

## Abstract

Stent implantation impairs local endothelial function and may be associated with subsequent adverse cardiovascular events. Telmisartan, an angiotensin II receptor blocker that has unique peroxisome proliferator-activated-receptor-gamma-mediated effects on cardiovascular disease, has been shown to enhance endothelial function and limit neointimal hyperplasia. This study utilized hybrid biodegradable/stent nanofibers to facilitate sustained and local delivery of telmisartan to injured arterial vessels. Telmisartan and poly(d,l)-lactide-*co*-glycolide (PLGA) (75:25) were dissolved in hexafluoroisopropyl alcohol and electrospun into biodegradable nanofibrous tubes which were coated onto metal stents. By releasing 20% of the loaded telmisartan in 30 days, these hybrid biodegradable/stent telmisartan-loaded nanofibers increased the migration of endothelial progenitor cells in vitro, promoted endothelialization, and reduced intimal hyperplasia. As such, this work provides insights into the use of PLGA nanofibers for treating patients with an increased risk of stent restenosis.

## 1. Introduction

Catheter-based interventional strategies represent the gold standard for revascularizing artery stenosis. However, balloon angioplasty or limus-eluting stent implantation result in damage to endothelial cells [1,2]. The functional vascular endothelium plays a critical role in maintaining the balance between blood coagulation, fibrinolysis, and vascular tone [3,4,5,6]. The loss of a continuous endothelial monolayer induces potentially lethal consequences such as thrombus formation and restenosis [7,8]. Delayed healing arises from several factors following percutaneous coronary intervention including device-related inflammation, endothelial dysfunction, and the development of neoatherosclerosis [9].

Telmisartan, an angiotensin II receptor blocker that is routinely used in clinical practice, can improve the function of endothelial cells, reduce responses to oxidative stress, and reduce superoxide-caused damage [10,11]. The drug also promotes endothelium-dependent relaxation, improves endothelial function [12,13], ameliorates the development and progression of atherosclerosis, and increases plaque stability [14,15].

This work develops biodegradable, electrospun hybrid poly(d,l)-lactide-*co*-glycolide (PLGA) nanofibers/stent for the local and prolonged delivery of telmisartan into a rabbit denuded artery. Electrospinning is a cost-effective and efficient process that produces continuous nanoscale fibers with diameters in the sub-micrometer to nanometer scale [16,17]. By using a high-voltage electric force, this technique generates loosely connected 3D porous mats with high porosity and large surface area. These characteristics allow the spun materials to emulate the structure of the extracellular matrix and, therefore, makes the nanofibers an excellent candidate for various medical applications [17,18]. PLGA is a Food and Drug Administration approved polymer used as an excipient for parenteral administrations [18,19]. PLGA is biocompatible and biodegradable, exhibits a wide range of degradation kinetics, and possesses tunable mechanical properties. Importantly, compared to other degradable biomaterials, PLGA has been extensively researched for the delivery of drugs, proteins, and other macromolecules [20]. In this manuscript, the characteristics of electrospun PLGA nanofibers are examined. Subsequently, the in vitro release patterns of telmisartan from nanofibers are characterized. Finally, the capacity of the nanofibers to promote endothelial recovery is explored.

## 2. Materials and Method

### 2.1. Fabrication of Telmisartan-Loaded Nanofibrous Tubes

PLGA (Resomer RG 756, Boehringer, Germany) has a lactide:glycolide ratio of 3:1 and a molecular weight of 76,000–115,000 Da. Telmisartan was purchased from Sigma-Aldrich (Saint Louis, MO, USA). Following our previous work [21,22], hexafluoroisopropyl alcohol (HFIP), also acquired from Sigma-Aldrich, was selected as the solvent. To perform the electrospinning process, a needle and syringe (0.42 mm), a metallic pin (0.95 mm) on a motor, a ground electrode, and a high voltage supply (35 kV and 4.16 mA/125 W) were set up [23]. Three hundred rpm was used for the rotational speed of the motor. A pre-determined weight ratio of PLGA to telmisartan (240 mg/40 mg, *w*/*w*) was dissolved into 1 mL of HFIP for the nanofiber electrospinning [24]. A syringe pump with a 3.6 mL/h volumetric flow rate and a metallic pin were then used to fabricate biodegradable nanofibrous tubes at room temperature. Subsequently, the biodegradable nanofibrous tubes were put together with a Gazella bare metal stent (3.0 × 14 mm, Biosensors International, Morges, Switzerland). A hybrid telmisartan nanofiber-loaded stent was thus obtained. All such stents were kept for 72 h to make the solvent evaporate.

### 2.2. Porosity

The porosity of the nanofibers was calculated using the following equation.
Porosity (%) = 1 − (ρ_nanofibers_/ρ_polymer_)(1)
where ρ_nanofibers_ and ρ_polymer_ denote the densities of the nanofibers and the polymer, respectively.

### 2.3. Scanning Electron Microscopy (SEM) Observation

Electrospun biodegradable nanofibers were first coated with gold and then characterized by an SEM. Size distribution of spun nanofibers was assayed from one hundred randomly chosen fibers (*n* = 3) utilizing ImageJ (National Institutes of Health, Bethesda, MD, USA).

### 2.4. Mechanical Properties of Materials

The mechanical properties of nanofibrous scaffolds were measured using a Lloyd tensiometer (AMETEK, Inc., Berwyn, PA, USA) with a 0.1 kN load cell using the ASTM D638 standard. A strip with an area of 50 mm by 10 mm from the scaffolds was used and gripped between two clamps 30 mm apart. The material was moved by the top clamp at 60 mm/min through a distance of 10 cm before the clamp was returned to its starting point. The elongation of the samples and the force on the scaffolds upon breaking of the scaffolds were recorded. The test was conducted four times (*n* = 4) on each scaffold. The elongation and tensile strength at breakage were obtained as follows.
Elongation at breakage (%) = Increase in length at breaking point (mm)/Original length (mm) × 100%(2)
Tensile strength (MPa) = Breaking force (N)/Cross-sectional area of sample (mm^2^)(3)

### 2.5. Wetting Angles

Using a water contact angle analyzer (First Ten Angstroms, Inc. Portsmouth, VA, USA), the wetting angles of the scaffolds were obtained. An area of 10 mm by 10 mm were prepared. Distilled water with a volume of 0.1 mL was lightly dropped onto the surfaces of the testing area and examined with a video monitor (*n* = 4).

### 2.6. Water Uptake Capacity

The water uptake capacities of the scaffolds were measured. The electrospun nanofibers were immersed in PBS and after 0.5, 1, 2, 3, 8, and 24 h the PBS on their surfaces was removed using filter paper and the nanofibers were weighed. The water content (WC, %) is calculated as follows.
WC (%) = (W − W_0_)/W_0_ × 100(4)
where W_0_ and W are the weights of the samples before and following immersion in PBS for the specified periods, respectively.

### 2.7. In Vitro Release

The in vitro release properties of telmisartan from the electrospun nanofibers were obtained using an elution method. The samples loaded with telmisartan were put in test tubes with 1 mL of PBS. Following a 24 h incubation at 37 °C, the eluent was analyzed. The PBS was replaced and the test tubes were incubated for another 24 h before the eluent was collected and analyzed. This process was repeated over 30 days.

Using a high-performance liquid chromatography (HPLC) assay, the telmisartan concentrations in the eluents were obtained. The HPLC analyses were performed using a Hitachi L-2200 Multisolvent Delivery System (Tokyo, Japan). A SYMMETRY C_8_, 3.9 cm × 150 mm HPLC column (Waters) was used to separate out for the separation of telmisartan. The mobile phase used for telmisartan was (0.05 M KH_2_PO_4_ + 1 mL phosphoric acid): acetonitrile 40:60 (*v*/*v*). The absorbency was monitored at a wavelength of 271 nm and the flow rate was 1.0 mL/min.

### 2.8. Cell Cultures for Migration Assay

The effect of the nanofiber-released telmisartan on endothelial progenitor cell (EPC) migration was then quantified [25,26,27,28,29]. The EPCs were a gift from the Laboratory of Molecular Pharmacology (Chang-Gung University, Taoyuan, Taiwan). EPCs were obtained by Ficoll-Hypaque (Sigma-Aldrich, Saint Louis, MO, USA) density-gradient centrifugation within 6 h of collection from peripheral mononuclear cells and were cocultured through a transwell with 8 μm pores (Corning Inc., Corning, NY, USA). The culture was maintained at 37 °C for 24 h in a humidified atmosphere that contained 5% CO_2_ to allow cell attachment. After a 24-h incubation, the medium was replaced with 600 μL Dulbecco’s modified Eagle medium (DMEM) (Gibco, Invitrogen, Waltham, MA, USA) that was supplemented with 2% (*v/v*) FBS (Gibco, Invitrogen), 100 U/mL penicillin, and 100 μg/mL streptomycin (Gibco, Invitrogen) and cells were cultured for another 24 h. On the following day, 6 × 10^4^ EPCs were seeded into the upper chamber of the transwell inserts. The culture was then maintained for two hours at 37 °C in a humidified atmosphere that contained 5% CO_2_. At the end of the process, the number of EPCs that had migrated to the other side of the transwell inserts was determined by hematoxylin staining. The EPCs that were cultured in transwells where the lower chambers contained cell-free DMEM served as controls.

### 2.9. In Vivo Study

Adult male New Zealand white rabbits with a mean mass of 3.22 ± 0.24 kg were used throughout the in vivo portions of this study. They were all housed in separate cages in a light- and temperature-controlled room and had free access to sterilized drinking water with standard rabbit chow ad libitum. In accordance with the regulations of the National Institute of Health of Taiwan, under the supervision of a licensed veterinarian, all animal procedures were institutionally approved and all of the studied animals were cared for (Chang Gung University CGU14-045).

Using a muscular injection of Zoletil 50 (tiletamine-zolazepam, 10 mg/kg) and xylazine (9.3 mg/kg), and oxygen (2 L/min) through a face mask, rabbits were anesthetized and sedated. Group A consisted of 12 rabbits with telmisartan-loaded hybrid stents; group B consisted of 12 rabbits with hybrid stents which did not contain telmisartan. The rabbits were treated with an endothelial denudation of the descending abdominal aorta using a 3.5 × 20 mm Maverick balloon (Boston Scientific, Maple Grove, MN, USA) to create an arterial balloon injury. After stent deployment, angiography was used to determine post-operative vessel patency. To satisfy institutional regulations and ethical concerns, aspirin (40 mg/d) was administered orally 24 h before catheterization and then continuously throughout the study. A single dose of intra-arterial heparin (150 IU/kg) was administered during the procedure.

Stented vessels were gathered at one month following deployment for microscopic observation and histological examination.

### 2.10. Microscopic Observation

Intact stented vessels were rinsed in 0.1 mmol/L sodium phosphate buffer (pH 7.2 ± 0.1) and subsequently post-fixed in 1% osmium tetroxide for approximately 30 min. They were then dehydrated in a graded series of ethanol solutions. All the tissue samples were mounted, sputter-coated with gold, and observed under an SEM (Hitachi S-3000N, Tokyo, Japan).

The degree of endothelialization of the stented area were estimated using low-power photographs of the lumen surface at a magnification of 35×. Increased magnification (200×) was then used to directly visualize the endothelial cells. The extent of the endothelial surface coverage on the stent struts was characterized by ImageJ (National Institutes of Health, Bethesda, MD, USA) [30].

### 2.11. Statistics and Data Analysis

All data are presented as mean ± standard deviation. One-way ANOVA was used to compare data to identify statistically significant relationships. Within ANOVA, the post hoc Bonferroni procedure for multiple comparisons was used to identify significant differences between pairs. Differences were regarded as statistically significant at *p* value < 0.05. Data were analyzed using SPSS software (version 17.0 for Windows; SPSS Inc, Chicago, IL, USA).

## 3. Results and Discussion

Hybrid biodegradable telmisartan-loaded nanofibers were developed using PLGA 756 and electrospinning. PLGA is a thoroughly-characterized, biodegradable, and biocompatible copolymer with medical applications that has been widely used in drug delivery systems [20,31]. Nanostructure PLGA drug-delivery systems with multiple-functionalities and diverse sizes, shapes, and bio-molecular conjugations are used to treat cardiovascular diseases [20,32]. Telmisartan, a partial agonist of peroxisome proliferator-activated receptor-gamma (PPARγ) [33], can improve the function of endothelial cells, reduce oxidative stress responses in the body, alleviate superoxide damage [34,35], and inhibit the growth factor-induced proliferation and migration of vascular smooth muscle cells. Several recent reports have suggested that telmisartan has anti-atherosclerotic effects, including improving sensitization to insulin, protection against weight gain, anti-inflammation, and the downsizing of adipocytes by PPARγ activation [36]. Jin et al. [13] studied the effect of telmisartan on coronary slow flow phenomenon (CSFP) and showed that telmisartan ameliorates endothelial dysfunction in CSFP. This study further locally delivered telmisartan to injured arterial vessels using stent/nanofibers, and investigated its impact on the vessels. Telmisartan-loaded PLGA 756 and plain PLGA 756 nanofibers were fabricated using electrospinning (Figure 1a). Gazella bare metal stents were prepared and telmisartan/PLGA 756 or PLGA 756 nanofibrous membranes were spun onto them to form hybrid stent/nanofibers (Figure 1b).

Figure 2a,b present SEM micrographs (magnification 3000×) of the electrospun nanofibrous membranes of telmisartan-loaded PLGA and plain PLGA, respectively. The diameters of the electrospun PLGA/telmisartan nanofibers and plain PLGA 756 were 720.3 ± 25.2 nm and 1289.5 ± 42.1 nm, respectively (*p* < 0.001, Figure 2c). In the electrospinning procedure, the polymeric mixture is stretched by the external electric force. With the addition of pharmaceuticals, the polymer content in the nanofibers decreased and became easier for the nanofibers to be extended by an external force. The diameter of the electrospun fibers decreased accordingly [37,38]. PLGA is a hydrophobic polyester copolymer of poly lactic acid (PLA) and poly glycolic acid (PGA) [39,40]. The experimental results in Figure 2d,e suggest that the water contact angle of the electrospun nanofibers decreased as the telmisartan loading (131.1 ± 4.4° vs. 109.6 ± 11.7°) increased (*p =* 0.041, Figure 2f). Taken together, it can be inferred that a mixture of water-soluble telmisartan and PLGA 756 reduced the amount of PLGA as a percentage of total volume, thus making the product more hydrophilic.

Cell-scaffold interaction is indirectly regulated by varying the wettability of the scaffold surfaces as more hydrophilic surfaces better accelerate cell adhesion and cell spreading [41,42]. As shown in Figure 2g, the water content of plain PLGA 756 nanofibers peaked at 60.2 ± 14.7% (vs. telmisartan group 56.3 ± 12.3%, *p* = 0.744) after a half-hour, while that of the telmisartan/PLGA 756 nanofibers peaked at 86.7 ± 18.4% at one hour (vs. plain PLGA group 20.8 ± 12.5%, *p* = 0.007). Thereafter, the water content in both groups declined with time (2 h later: 73.7 ± 22.9% vs. 8.0 ± 1.7%, *p* = 0.037; 3 h later: 74.6 ± 18.1% vs. 3.0 ± 1.6%, *p* = 0.020; 8 h later: 57.4 ± 12.2% vs. 13.4 ± 0.5%, *p* = 0.003; 24 h later: 30.6 ± 10.7% vs. 1.5 ± 1.6%, *p* = 0.040, in telmisartan and plain PLGA 756 groups, respectively). Additionally, the porosity of the nanofibrous membranes in the telmisartan group was higher (76.4 ± 5.0%) than that in the plain PLGA756 group (58.4 ± 6.6%) (*p* = 0.005) (Figure 2h). High porosity favors both the efficient influx of nutrients and the outflow of catabolic waste when cells are seeded on a scaffold [43]. It follows that porous scaffolds with hydrophilic surfaces should promote cellular adhesion, proliferation, and, possibly, enhanced functionality.

In aqueous environments, through bulk or heterogeneous erosion, PLGA is degraded by hydrolysis of its ester linkages. A compound’s release rate depends on the properties of the polymer and drugs in the physiological environment [44,45]. In general, drug release from a drug-loaded resorbable device occurs over three distinct stages: burst release, diffusion-dominated elution, and degradation-dominated release. During electrospinning, most drugs are encapsulated in the volume of the PLGA matrix. Nonetheless, a few molecules may be located on nanofibers’ surfaces, thus resulting in the initial burst. After the burst, the drug-release curve is controlled by both the diffusion effect and the polymer degradation [39,46]. The in vitro release behavior of the hybrid telmisartan/PLGA 756 nanofibers was determined using an elution method and an HPLC assay. The accumulated release curves suggest that the hybrid stents continuously released telmisartan over 30 days (8.9% of the telmisartan was released by day 3, 12.6% by day 7, and 17.9% by day 20). A tri-phasic drug release profile [40] was thus observed for the drug-loaded 75:25 PLGA nanofibers, namely, a burst release at day 1, a fast diffusion release at days 2–5, and thereafter a gradually diminishing release (Figure 3a).

Optimal stent implantation requires high inflation pressures and high-pressure angioplasty, defined as dilation at a pressure of 18 atm [47] (approximately 1.82 MPa) to ensure complete balloon and stent expansion. With respect to the mechanical properties of the electrospun nanofibers, telmisartan/PLGA 756 nanofibers had a lower tensile strength than the plain PLGA 756 fibers (4.1 ± 0.3 vs. 5.1 ± 0.4 MPa, respectively) (*p* = 0.010) (Figure 3b). Additionally, the PLGA 756 nanofibers exhibited greater elongation at breakage (314.2 ± 13.1%) than the plain PLGA group (121.5 ± 12.0%) (*p* < 0.001). As such, the two nanofibrous membranes have adequate properties for stent expansion stress tolerance.

Endothelial progenitor cells (EPCs) form a circulating pool of cells that can create a cellular patch at sites of endothelial injury, contributing directly to the homeostasis and repair of the endothelial layer. Virtually all risk factors for atherosclerosis have been associated with a decrease in either the abundance, the dysfunction, or both, of circulating EPCs [48,49]. Telmisartan ameliorates the cellular senescence of EPCs by reducing oxidative stress and maintains endothelial function by reducing the production of reactive oxygen species, inducing the secretion of inflammatory cytokines from endothelial cells, and deactivating apoptotic signaling pathways [50,51]. In clinical trials, the inhibition of oxidative stress by telmisartan correlated with improvement in EPC numbers and function [52,53]. An EPC migration study (Figure 4a–d) revealed significantly more migration of cells on day 3 (145.7 ± 2.5%), 7 (133.8 ± 3.3%), and 28 (125.2 ± 3.3%) with the eluent compared to a control of PBS alone (Figure 4e) (all *p* < 0.001).

Figure 5 characterizes the re-endothelialization of damaged arterial walls. The surfaces of the struts were almost entirely covered with regularly shaped endothelial cells in close contact with each other (Figure 5a,b). However, poor alignment and extended irregular intercellular spaces were identified under higher magnification (200×) (Figure 5c,d). Additionally, significant intimal hyperplasia was observed in the control group (Figure 6e,f). The endothelia-dependent vasodilation in reaction to Ach (0.5 μg/mL/min) was considerably better in the telmisartan-loaded stents (4.8 ± 2.2) than in the non-telmisartan-loaded (−1.9 ± 1.5) stents (*p* = 0.002) (Figure 5g). After 28 days, endothelial coverage on the surfaces of struts revealed that re-endothelialization was significantly greater in the telmisartan group (99.3 ± 0.1%) compared to the control group (95.3 ± 1.0%) (*p* = 0.001) (Figure 5h).

The relative ratio of HO-1 around the stenting area at four weeks post-surgery is shown in Figure 6a. The HO-1/GAPDH ratios in telmisartan/PLGA 756 group B (0.29 ± 0.02) were greater than that in the PLGA 756 group (0.18 ± 0.03) (Figure 6b) (*p* < 0.001). Heme oxygenase-1 (HO-1) plays a critical role in the prevention of vascular inflammation and may protect against atherosclerosis by preserving vascular cell function and survival [54]. Endothelial cell dysfunction that manifests as impaired endothelium-dependent vasodilation and endothelial nitric oxide (NO) synthesis is one of the earliest changes associated with the development of atherosclerosis. The overexpression of HO-1 improves endothelium-dependent vascular relaxation and restores endothelial NO synthase expression in various animal models [55]. Interestingly, 28 days following implantation, the stents that were loaded with telmisartan were associated with a higher local expression of HO-1 than the non-loaded stents in the control group (*p* < 0.001). Therefore, telmisartan may induce an upregulation of HO-1 which blocks the inflammatory response in vascular cells and potently inhibits vascular smooth muscle cell proliferation.

## 4. Conclusions

Electrospun hybrid biodegradable/stent telmisartan-loaded nanofibers significantly improved various metrics pertinent to vessel revascularization. In vitro and in vivo analyses of hybrid stent/PLGA 756 nanofibers with telmisartan loading indicate that they can controllably release disease-relevant therapeutics, enhance the migration of EPCs, provide complete endothelial coverage and recovery, and reduce intimal hyperplasia. Taken together, these findings suggest that hybrid stents/telmisartan-loaded nanofibers have widespread potential applications in cardiovascular drug delivery. Specifically, the use of hybrid nanofibers with the capacity for sustained and local drug delivery may prove particularly useful in the treatment of patients with an increased risk of arterial restenosis.

## Figures and Tables

**Figure 1 pharmaceutics-13-01756-f001:**
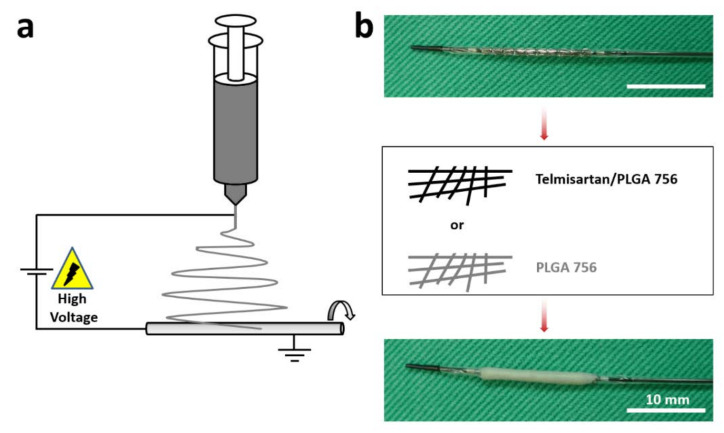
(**a**) Electrospinning schematic. (**b**) Entire coating process. Coating a bare metal stent (3.0 × 14 mm) with either Telmisartan/PLGA 756 or PLGA 756 nanofibers.

**Figure 2 pharmaceutics-13-01756-f002:**
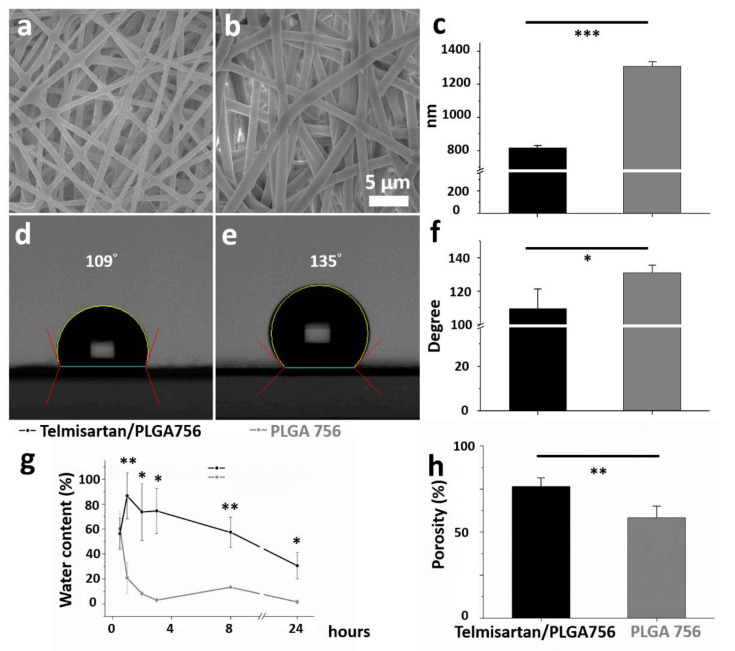
(**a**) Blended nanofibrous membranes of telmisartan/PLGA 756 and (**b**) plain PLGA 756. (**c**) Mean fiber diameters in both nanofiber groups. (**d**) Water contact angles of telmisartan/PLGA 756 and (**e**) plain PLGA 756. (**f**) Mean contact angles in both groups. (**g**) Difference between water contents of two nanofibers after 24 h of PBS immersion and (**h**) their porosities. (*** *p* < 0.001; ** *p* < 0.01; * *p* < 0.05).

**Figure 3 pharmaceutics-13-01756-f003:**
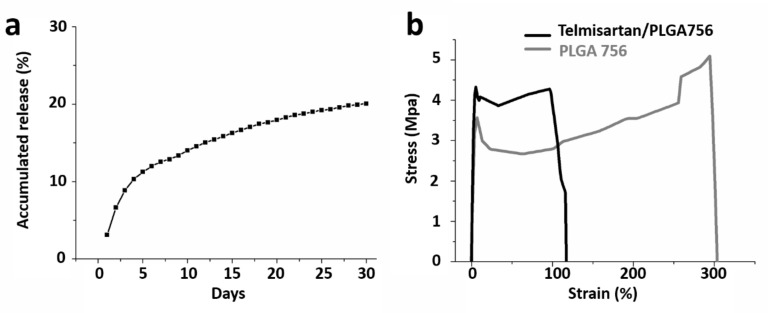
(**a**) Accumulated release of telmisartan and (**b**) stress–strain curves of telmisartan-loaded and plain PLGA 756 nanofibrous membranes.

**Figure 4 pharmaceutics-13-01756-f004:**
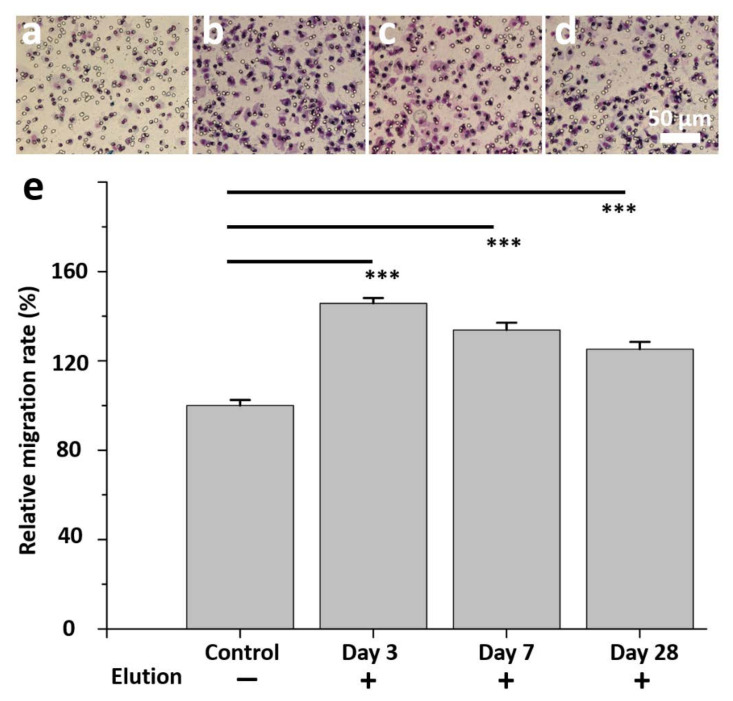
In vitro study of eluent of telmisartan/PLGA 756 scaffolds to test endothelial progenitor cells migration for two hours. (**a**) Control group without eluent. (**b**) EPCs with day 3 eluent. (**c**) EPCs with day 7 eluent. (**d**) EPCs with day 28 eluent. (**e**) Migration of cells in control group is significantly less than telmisartan/PLGA 756 scaffolds group. (*** *p* < 0.001).

**Figure 5 pharmaceutics-13-01756-f005:**
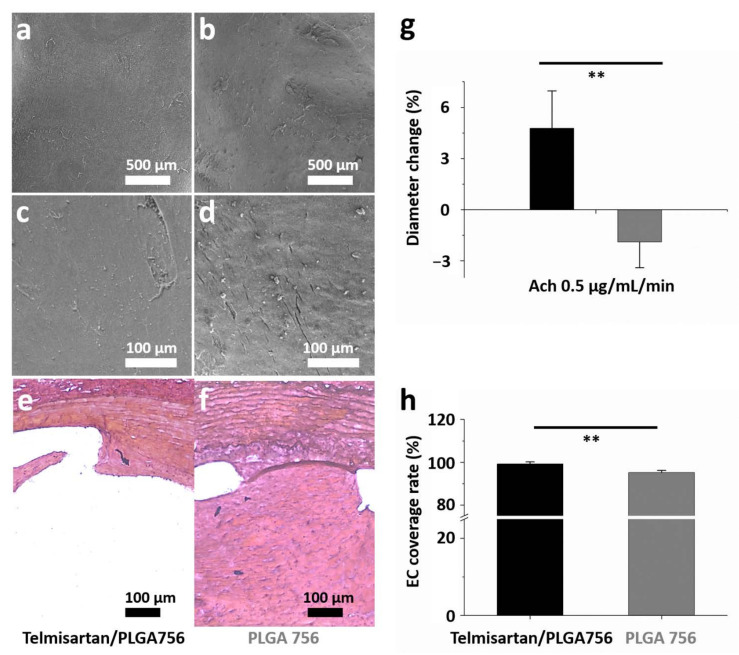
On day 28, the data were collected for ((**a**,**b**), 35×; (**c**,**d**), 200×) morphology of endothelium under various magnifications and (**e**,**f**) response of intimal hyperplasia after four weeks of stenting treatment. (**g**) In vivo endothelial function was evaluated from variation in abdominal aorta diameter using acetylcholine (Ach) infusions. (**h**) Endothelium coverage in different groups. (** *p* < 0.01).

**Figure 6 pharmaceutics-13-01756-f006:**
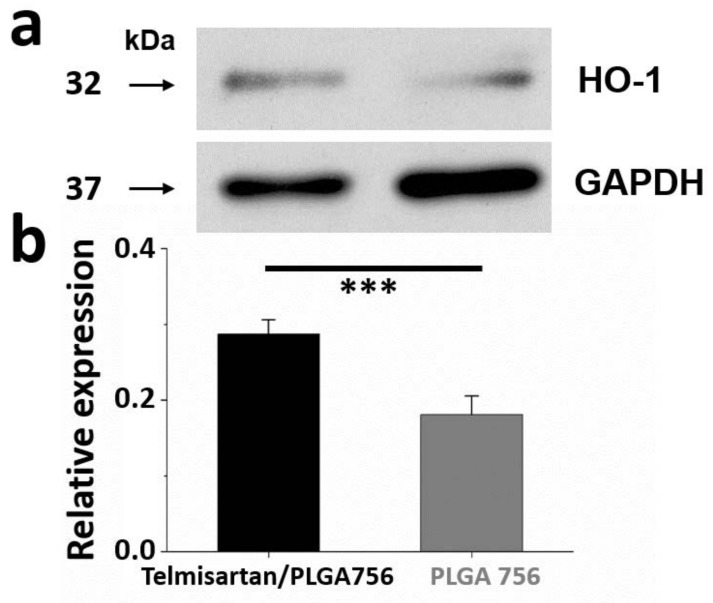
(**a**) Western blot of HO-1 content. (**b**) Relative expression was quantified using densitometry as ratio of density to GAPDH. (*** *p* < 0.001).

## Data Availability

The data used to support the findings of this study are available from the corresponding author upon request.

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
