# Peer review of "Telmisartan Loaded Nanofibers Enhance Re-Endothelialization and Inhibit Neointimal Hyperplasia"

_pharmaceutics, 2021, doi:10.3390/pharmaceutics13111756_

Round 1

Reviewer 1 Report

The submitted article presents the local delivery of telmisartan, an angiotensin II receptor blocker, from fibers on the basis of poly(D,L)-lactide-co-glycolide (PLGA) (75:25). The obtained electrospun fibers were implanted into a metal stent. The release of telmisartan was 20% for the duration of 30 days. The drug-loaded fibers increased the migration of endothelial progenitor cells in vitro, promoted endothelialization, and reduced intimal hyperplasia.

Specific points requiring attention are detailed below.

Comment 1. The introduction is very brief. It does not present the advantages of electrospinning process which resulted in preparation of nano- and microfibers? The introduction should include also why the authors choose to work with the copolymer of polylactide: which are its advantages, what is known in the literature, etc.?

Comment 2. In the Materials section: the authors have to include the molecular weight characteristics of the copolymer.

Comment 3. Why hexafluoroisopropyl alcohol is used as solvent?

Comment 4. In the wetting angles measurements: the volume of the water drop is not presented.

Comment 5. For the in vitro release: At which wavelength is the absorbense recorded?

Comment 6. The definition used by authors that the fibers obtained by them are nanofibers is not correct. The presented fiber diameters are 817.7±12.6 nm and 1311.0±24.8 nm for the PLGA/telmisartan nano-fibers and plain PLGA 756, respectively. The obtained fibers are in micrometer range.

Comment 7. Figure 1 A is a schematic presentation of electrospinning that is used from decades and do not present the used by the authors electrospinning set-up (they have used a rotating drum).

Comment 8. Figure 2: By magnifying the presented SEM micrographs with the marker of 5 μm it is easily seen that the fiber diameters are much thicker that the given by the authors.

The authors dealt with a problem that has been known for decades, the incorporation of drugs into the PLA carrier by electrospinning. This is not a new procedure at all. The introduction is brief and inexhaustible. The molecular characteristics of PLGA are missing. The fibers are thicker than 1311.0±24.8 nm. The standard deviation of fiber diameters is also higher than the given one. The novelty, quality and presentation are insufficient.That is why I recommend that the manuscript is not accepted for publication in Pharmaceutics.

Author Response

REVISIONS COMPLETED IN RESPONSE TO THE REVIEWERS’ COMMENTS AND SUGGESTIONS

We thank the Editor and the reviewers for their careful read and constructive criticisms on previous draft. We have carefully taken the comments into consideration in preparing our revision; the following summarizes how we responded to the comments.

Reviewer 1

Comment 1. The introduction is very brief. It does not present the advantages of electrospinning process which resulted in preparation of nano- and microfibers? The introduction should include also why the authors choose to work with the copolymer of polylactide: which are its advantages, what is known in the literature, etc.?

Response: Thank you for the comments. Electrospinning is a cost-effective and efficient process that produces continuous nanoscale fibers with diameters in the sub-micrometer to nanometer scale.16, 17 By using a high-voltage electric force, this technique generates loosely connected 3D porous mats with high porosity and large surface area. These characteristics allow the spun materials to emulate the structure of the extracellular matrix structure and, therefore, makes the nanofibers an excellent candidate for various medical applications19,20. PLGA is a Food and Drug Administration approved polymer used as an excipient for parenteral administrations.18, 19 PLGA is biocompatible and biodegradable, exhibits a wide range of degradation kinetics, and possesses tunable mechanical properties. Importantly, compared to other degradable biomaterials, PLGA has been extensively researched for the delivery of drugs, proteins, and other macromolecules.20 The Introduction section has been revised to better explain the advantages of electrospinning and PLGA material (page 2).

Comment 2. In the Materials section: the authors have to include the molecular weight characteristics of the copolymer.

Response: Thank you for the comments. PLGA (Resomer RG 756, Boehringer, Germany) has a lactide:glycolide ratio of 3:1 and a molecular weight of 76,000-115,000 Da. The manuscript has been revised to provide the molecular weight of PLGA (page 2).

Comment 3. Why hexafluoroisopropyl alcohol is used as solvent?

Response: Thank you for the comments. Following our previous work21, 22, hexafluoroisopropyl alcohol (HFIP), also acquired from Sigma-Aldrich, was selected as the solvent. The manuscript has been revised to provide better explanation (page 2). Meanwhile, the relevant references [21,22] have also been added in the reference list.

Comment 4. In the wetting angles measurements: the volume of the water drop is not presented.

Response: Thank you for the comments. Distilled water with a volume of 0.1 mL was lightly dropped onto the surfaces of the testing area and examined with a video monitor (n=4). The text has been revised to provide the volume of the water drop used in the experiments (page 3).

Comment 5. For the in vitro release: At which wavelength is the absorbense recorded?

Response: Thank you for the comments. The mobile phase used for telmisartan was (0.05 M KH2PO4+1 mL phosphoric acid): acetonitrile 40:60 (v/v). The absorbency was monitored at a wavelength of 271 nm and the flow rate was 1.0 mL/min. The manuscript has been revised to provide the detailed information for the HPLC assay (page 3).

Comment 6. The definition used by authors that the fibers obtained by them are nanofibers is not correct. The presented fiber diameters are 817.7±12.6 nm and 1311.0±24.8 nm for the PLGA/telmisartan nano-fibers and plain PLGA 756, respectively. The obtained fibers are in micrometer range.

Response: Thank you for the comments. Electrospun fibers ranged from a few hundred nanometers to more than a thousand nanometers.

Comment 7. Figure 1 A is a schematic presentation of electrospinning that is used from decades and do not present the used by the authors electrospinning set-up (they have used a rotating drum).

Response: Thank you for the comments. Figure 1A has been redrawn to better reflect the electrospinning technique (rotating drum) used in this study.

Comment 8. Figure 2: By magnifying the presented SEM micrographs with the marker of 5 μm it is easily seen that the fiber diameters are much thicker that the given by the authors.

Response: Thank you for the comments. We are sorry for the mis-marking of the scale bar. The scale bar in Figure 2 has been re-marked.

The authors dealt with a problem that has been known for decades, the incorporation of drugs into the PLA carrier by electrospinning. This is not a new procedure at all. The introduction is brief and inexhaustible. The molecular characteristics of PLGA are missing. The fibers are thicker than 1311.0±24.8 nm. The standard deviation of fiber diameters is also higher than the given one. The novelty, quality and presentation are insufficient. That is why I recommend that the manuscript is not accepted for publication in Pharmaceutics.

Response: Thank you for the comments. The manuscript has been revised accordingly to provide the additional information.

Reviewer 2 Report

The manuscript presents a hybrid stent covered with a nanofibrous PLGA releasing telmisartan. Many papers cover this topic and one where authors used the same drug with PCL/gelatin nanofibers for vascular scaffolds.    - What is the solubility of telmisartan in HFIP? - Did you measure viscosity decrease after adding such a small amount of drug? In my opinion, other issues could lead to fibre diameter reduction.  - What is the solubility of telmisartan in PBS. Few papers state that it is extremely low soluble in water (BCS II). Can it affect the release study? - In one of the paragraphs, you mention PLGA degradation and conclude with the drug release from PLGA nanofibers. What is PLGA (RG756) degradation time? It is most probably several months, not three weeks. Therefore we can assume that it does not affect drug release at such an early stage. - Why do you state that the release pattern shows biphasic drug release? It is the opposite. I don't know whether it is fully solubilized in the PLGA matrix, but if it is, then at the beginning, we observe the release of drug from the surfaces (pores) easily contacted with buffer.  - Did you inflate the balloon with stent and fibrous covering? Did it break?  

Author Response

REVISIONS COMPLETED IN RESPONSE TO THE REVIEWERS’ COMMENTS AND SUGGESTIONS

We thank the Editor and the reviewers for their careful read and constructive criticisms on previous draft. We have carefully taken the comments into consideration in preparing our revision; the following summarizes how we responded to the comments.

Reviewer 2

The manuscript presents a hybrid stent covered with a nanofibrous PLGA releasing telmisartan. Many papers cover this topic and one where authors used the same drug with PCL/gelatin nanofibers for vascular scaffolds. 

- What is the solubility of telmisartan in HFIP? - Did you measure viscosity decrease after adding such a small amount of drug? In my opinion, other issues could lead to fibre diameter reduction. 

Response: Thank you for the comments. Telmisartan was well dissolved in HFIP for the subsequent electrospinning experiments. In the electrospinning procedure, the polymeric mixture is stretched by the external electric force. With the addition of pharmaceuticals, the polymer content in the nanofibers decreased. It becomes easier for the nanofibers to be extended by the exterior force. The sizes of the electrospun fibers decreased accordingly (page 5).

- What is the solubility of telmisartan in PBS. Few papers state that it is extremely low soluble in water (BCS II). Can it affect the release study?

Response: Thank you for the comments. The solubility of telmisartan is PBS is low. This might explain why the nanofibers only released about 20% of loaded drugs after 30 days of elution in PBS.

- In one of the paragraphs, you mention PLGA degradation and conclude with the drug release from PLGA nanofibers. What is PLGA (RG756) degradation time? It is most probably several months, not three weeks. Therefore we can assume that it does not affect drug release at such an early stage.

Response: Thank you for the comments. The degradation time of PLGA ranges from 1-3 months. In general, drug release from a drug-loaded resorbable device occurs over three distinct stages: burst release, diffusion-dominated elution, and degradation-dominated release. During electrospinning, most drugs are encapsulated in the volume of the PLGA matrix. Nonetheless, a few molecules may be located on nanofibers’ surfaces, thus resulting in the initial burst. After the burst, the drug-release curve is both controlled by the diffusion effect and the polymer degradation.39, 46 The in vitro release behavior of the hybrid telmisartan/PLGA 756 nanofibers was determined using an elution method and an HPLC assay. The accumulated release curves suggest that the hybrid stents continuously released telmisartan over 30 days (8.9 % of the telmisartan was released by day 3, 12.6% by day 7, and 17.9 % by day 20). A tri-phasic drug release profile40 was thus observed for the drug-loaded 75:25 PLGA nanofibers, namely a burst release at day 1, a fast diffusion release at days 2-5, and thereafter a gradually diminishing release (Fig. 3a). The manuscript has been revised to better explain the drug release characteristics (page 5).

- Why do you state that the release pattern shows biphasic drug release? It is the opposite. I don't know whether it is fully solubilized in the PLGA matrix, but if it is, then at the beginning, we observe the release of drug from the surfaces (pores) easily contacted with buffer. 

Response: Thank you for the comments. In general, drug release from a drug-loaded resorbable device occurs over three distinct stages: burst release, diffusion-dominated elution, and degradation-dominated release. During electrospinning, most drugs are encapsulated in the volume of the PLGA matrix. Nonetheless, a few molecules may be located on nanofibers’ surfaces, thus resulting in the initial burst. After the burst, the drug-release curve is both controlled by the diffusion effect and the polymer degradation.39, 46 The in vitro release behavior of the hybrid telmisartan/PLGA 756 nanofibers was determined using an elution method and an HPLC assay. The accumulated release curves suggest that the hybrid stents continuously released telmisartan over 30 days (8.9 % of the telmisartan was released by day 3, 12.6% by day 7, and 17.9 % by day 20). A tri-phasic drug release profile40 was thus observed for the drug-loaded 75:25 PLGA nanofibers, namely a burst release at day 1, a fast diffusion release at days 2-5, and thereafter a gradually diminishing release (Fig. 3a). The manuscript has been revised to better explain the drug release characteristics (page 5).

- Did you inflate the balloon with stent and fibrous covering? Did it break?  

Response: Thank you for the comments. Yes, the nanofibrous membranes might be broken after being inflated by the balloon. Nevertheless, the membranes remained intact in between the metallic stent and the blood vessels after deployment.

Reviewer 3 Report

Research article submitted to Pharmaceutics “Local Delivery of Telmisartan Nanofibers Enhances Re-Endo-thelialization and Inhibits Neointimal Hyperplasia” is no doubt an interesting work but presentation of this article is not suitable enough to be considered for publication in its current form. I would like to request authors to revise their manuscript and resubmit it. There are few suggestions for authors which may help authors in revision of the manuscript.

  1. Title is not clear. It may be written as “Telmisartan Loaded Nanofibers Enhance Re-Endo-thelialization and Inhibits Neointimal Hyperplasia” or “Controlled/Local Delivery of Telmisartan Enhances Re-Endo-thelialization and Inhibits Neointimal Hyperplasia”. These are just as reference, authors can write whatever they think will be better for readers to understand their work.
  2. Authors have tried to write this manuscript as complex as possible, but most of readers need simple but precise English to understand the work. It seems authors have used synonyms of many words throughout text which do not make any sense as per context. Authors are advised to proofread and rewrite sections in precise. However, methodology section is written well and is understandable.
  3. Poly(D,L)-lactide-co-glycolide (PLGA) is no doubt an excellent candidate for medical applications, however there are several polymers (PLA, PHBH, PCL etc) which are also widely used in medical applications. Did authors compare PLGA with any other polymer? How did they select PLGA for this specific application? Need explanation, please add this information in last paragraph of introduction section as well.
  4. Authors have cited an article “Jin Z, Tan Q and Sun B. Telmisartan ameliorates vascular endothelial dysfunction in coronary slow flow phenomenon (CSFP). Cell biochemistry and function. 2018;36:18-26” which shows that Telmisartan was already used in similar application (endothelial dysfunction), so what was significance of already proven drug to be used for this study?
  5. In SEM methods it was written as “For SEM analysis, the electrospun biodegradable nanofibers were covered with gold and then characterized. One hundred randomly selected samples were obtained using Image J software for diameter analysis (National Institutes of Health, Bethesda, MD, USA)”, please rewrite this paragraph so that it can be understandable for readers. It seems that authors tried to avoid any plagiarism and rearranged the sentence structures.
  6. Authors used endothelial progenitor cell (EPC) for in-vitro cell study. Authors should also write type (batch/code details) of cell lines used in this experiment. Further, study some relevant literature on in-vitro cell study. Here are some relevant examples to follow: https://doi.org/10.2147/IJN.S197665 ; https://doi.org/10.1021/acsanm.0c01562 ; https://doi.org/10.1016/j.mtcomm.2020.101161 ; https://doi.org/10.1016/j.ijbiomac.2020.03.237 ; https://doi.org/10.1038/s41598-019-49132-x

Author Response

REVISIONS COMPLETED IN RESPONSE TO THE REVIEWERS’ COMMENTS AND SUGGESTIONS

We thank the Editor and the reviewers for their careful read and constructive criticisms on previous draft. We have carefully taken the comments into consideration in preparing our revision; the following summarizes how we responded to the comments.

Reviewer 3

Research article submitted to Pharmaceutics “Local Delivery of Telmisartan Nanofibers Enhances Re-Endo-thelialization and Inhibits Neointimal Hyperplasia” is no doubt an interesting work but presentation of this article is not suitable enough to be considered for publication in its current form. I would like to request authors to revise their manuscript and resubmit it. There are few suggestions for authors which may help authors in revision of the manuscript.

  1. Title is not clear. It may be written as “Telmisartan Loaded Nanofibers Enhance Re-Endo-thelialization and Inhibits Neointimal Hyperplasia” or “Controlled/Local Delivery of Telmisartan Enhances Re-Endo-thelialization and Inhibits Neointimal Hyperplasia”. These are just as reference, authors can write whatever they think will be better for readers to understand their work.

Response: Thank you for the comments. The title of the paper has been revised to “Telmisartan Loaded Nanofibers Enhances Re-Endothelialization and Inhibits Neointimal Hyperplasia”.

  1. Authors have tried to write this manuscript as complex as possible, but most of readers need simple but precise English to understand the work. It seems authors have used synonyms of many words throughout text which do not make any sense as per context. Authors are advised to proofread and rewrite sections in precise. However, methodology section is written well and is understandable.

Response: Thank you for the comments. The manuscript has been proofed again by an English-speaking editor to improve tis readability.

  1. Poly(D,L)-lactide-co-glycolide (PLGA) is no doubt an excellent candidate for medical applications, however there are several polymers (PLA, PHBH, PCL etc) which are also widely used in medical applications. Did authors compare PLGA with any other polymer? How did they select PLGA for this specific application? Need explanation, please add this information in last paragraph of introduction section as well.

Response: Thank you for the comments. PLGA is a Food and Drug Administration approved polymer used as an excipient for parenteral administrations.18, 19 PLGA is biocompatible and biodegradable, exhibits a wide range of degradation kinetics, and possesses tunable mechanical properties. Importantly, compared to other degradable biomaterials, PLGA has been extensively researched for the delivery of drugs, proteins, and other macromolecules.20 In this manuscript, the characteristics of electrospun PLGA nanofibers are examined. Subsequently, the in vitro release patterns of telmisartan from nanofibers are characterized. Finally, capacity of the nanofibers to promote endothelium recovery is explored. The manuscript has been revised to better address the advantages of PLGA materials (page 2).

  1. Authors have cited an article “Jin Z, Tan Q and Sun B. Telmisartan ameliorates vascular endothelial dysfunction in coronary slow flow phenomenon (CSFP). Cell biochemistry and function. 2018;36:18-26” which shows that Telmisartan was already used in similar application (endothelial dysfunction), so what was significance of already proven drug to be used for this study?

Response: Thank you for the comments. Jin et al.13 studied the effect of telmisartan on coronary slow flow phenomenon (CSFP) and showed that that telmisartan ameliorates endothelial dysfunction in CSFP. This study further locally delivered telmisartan to injured arterial vessels using stent/nanofibers, and investigated its influence on the vessels. The manuscript has been revised to better address the work completed in this study (page 5).

  1. In SEM methods it was written as “For SEM analysis, the electrospun biodegradable nanofibers were covered with gold and then characterized. One hundred randomly selected samples were obtained using Image J software for diameter analysis (National Institutes of Health, Bethesda, MD, USA)”, please rewrite this paragraph so that it can be understandable for readers. It seems that authors tried to avoid any plagiarism and rearranged the sentence structures.

Response: Thank you for the comments. Electrospun biodegradable nanofibers were first coated with gold and then characterized by a SEM. Size distribution of spun nanofibers was assayed from one hundred randomly chosen fibers (N=3) utilizing Image J (National Institutes of Health, Bethesda, MD, USA). The entire sub-section has been revised to improve its readability (page 2).

  1. Authors used endothelial progenitor cell (EPC) for in-vitro cell study. Authors should also write type (batch/code details) of cell lines used in this experiment. Further, study some relevant literature on in-vitro cell study. Here are some relevant examples to follow: https://doi.org/10.2147/IJN.S197665 ; https://doi.org/10.1021/acsanm.0c01562 ; https://doi.org/10.1016/j.mtcomm.2020.101161 ; https://doi.org/10.1016/j.ijbiomac.2020.03.237 ; https://doi.org/10.1038/s41598-019-49132-x

Response: Thank you for the comments. The effect of the nanofiber-released telmisartan on endothelial progenitor cell (EPC) migration was then quantified.25-29 The EPCs were a gift from the Laboratory of molecular pharmacology (Chang-Gung University, Taiwan). EPCs were obtained by Ficoll-Hypaque (Sigma) density-gradient centrifugation within 6 hours of collection from peripheral mononuclear cells and were cocultured through a transwell with 8 μm pores (Corning, USA). The manuscript has been revised to provide information of cell lines used in the experiment (page 3). Relevant literature [Ref. 25-29] have been added in the reference list and cited in the text.

Round 2

Reviewer 1 Report

The authors have to determine more precisely the fiber diameters and the fibers`standart deviations. All the other comments have been addressed properly. I recommend that the manuscript is accepted after minor revisions for publication in Pharmaceutics.

Author Response

Reviewer #1

The authors have to determine more precisely the fiber diameters and the fibers` standart deviations. All the other comments have been addressed properly. I recommend that the manuscript is accepted after minor revisions for publication in Pharmaceutics.

Response: Thank you for the comments. We have re-evaluated the fiber diameters and updated the information in the revised manuscript (page 5).

Reviewer #2

The authors did respond to Reviewer questions and I don't have further questions. However, the quality of the paper is low.

Response: Thank you for the comments. One of the co-authors, Julien G. Roth who is a native English-speaking person, has proofread again the entire manuscript to improve the quality of the paper.

Reviewer 2 Report

The authors did respond to Reviewer questions and I don't have further questions. However, the quality of the paper is low.

Author Response

(The authors gave the same response as above.)
